# Water Filtration Membranes Based on Non-Woven Cellulose Fabrics: Effect of Nanopolysaccharide Coatings on Selective Particle Rejection, Antifouling, and Antibacterial Properties

**DOI:** 10.3390/nano11071752

**Published:** 2021-07-05

**Authors:** Blanca Jalvo, Andrea Aguilar-Sanchez, Maria-Ximena Ruiz-Caldas, Aji P. Mathew

**Affiliations:** Department of Materials and Environmental Chemistry, Stockholm University, Frescativägen 8, 10691 Stockholm, Sweden; Blanca.Jalvo@aces.su.se (B.J.); andrea.aguilar@mmk.su.se (A.A.-S.); mariaximena.ruizcaldas@mmk.su.se (M.-X.R.-C.)

**Keywords:** non-woven, cellulose nanocrystals, chitin nanocrystals, TEMPO-oxidized cellulose nanofibers, rejection, antifouling, antibacterial, membrane, coating

## Abstract

This article presents a comparative study of the surface characteristics and water purification performance of commercially available cellulose nonwoven fabrics modified, via cast coating, with different nano-dimensioned bio-based carbohydrate polymers, viz. cellulose nanocrystals (CNC), TEMPO-oxidized cellulose nanofibers (T-CNF), and chitin nanocrystals (ChNC). The surface-modified nonwoven fabrics showed an improvement in wettability, surface charge modification, and a slight decrease of maximum pore size. The modification improved the water permeance in most of the cases, enhanced the particle separation performance in a wide range of sizes, upgraded the mechanical properties in dry conditions, and showed abiotic antifouling capability against proteins. In addition, T-CNF and ChNC coatings proved to be harmful to the bacteria colonizing on the membranes. This simple surface impregnation approach based on green nanotechnology resulted in highly efficient and fully bio-based high-flux water filtration membranes based on commercially available nonwoven fabrics, with distinct performance for particle rejection, antifouling and antibacterial properties.

## 1. Introduction

Membranes and filters are routinely used in water and wastewater treatment processes due to their reliable removal of contaminants without the addition of new chemicals and the production of harmful products in water treatment processes. The removal efficiency or selectivity of the membrane/filters is related to its microstructure, pore size, and surface chemistry. Nano (0.1–100 nm range), micro (0.1–100 µm), and milli (pore size 0.1–100 mm) pores based on International Union of Pure and Applied Chemistry (IUPAC) definition allows the progressive removal of multivalent ions, virus, bacteria, and suspended particles by membranes and filters [1]. The pressure required for the separation process progressively decreases with an increase in pore size. Microfiltration (MF) membranes and filters are extensively used in wastewater treatment as a key step in the primary disinfection of the uptake water stream. The use of MF membranes presents a physical means of separation (a barrier) as opposed to a chemical alternative. In this sense, both filtration and disinfection take place in a single step, negating the extra cost of chemical dosage and the corresponding equipment. Moreover, MF membranes are used in secondary wastewater effluents to remove turbidity but also to provide treatment for disinfection.

As one of the fastest-growing segments, nonwovens have evolved to become the media of choice for MF [2]. Nonwoven fabric filters are one of the four major filtration systems in addition to woven, paper, and membrane filters in the market, and they can be engineered very precisely to meet exact specifications and stringent regulatory requirements for air and liquid filtration. Nonwovens offer many unique technical characteristics, including greater permeability, greater specific surface area, and controllable pore size distribution; they have distinct different filtration mechanisms and advantages for greater filtration efficiency, lower energy consumption, and better foulant-cake discharge properties [3] hence, longer service life. Together with this rise of nonwoven fabric usage, more and more nanofiber-based products are finding applications either as stand-alone filter media or in combination with conventional nonwovens.

Green bio-nanomaterials, such as cellulose nanocrystals (CNC), cellulose nanofibers (CNF), TEMPO-oxidized cellulose nanofibrils (T-CNF), chitin nanocrystals (ChNC), and their combinations possess high surface activity, surface charge, and surface area required to tailor demanding features such as nanostructured and nano-enabled bio-based membranes [4,5]. These nanomaterials also have good film formation ability [6] (a highly required feature in coating and surface treatment applications) and certain antifouling activity [7,8,9]. Previous studies of our group revealed the outstanding performance of nanocellulose and nanochitin as functional materials in the next generation affinity membranes for water purification [5,7,9,10,11,12,13].

We have earlier reported fully bio-based, superhydrophilic, high flux water filtration filters based on cellulose acetate electrospun membranes impregnated and nanotextured with CNCs or ChNCs [7,12]. The ability of these polysaccharide nanomaterials to form nano-scaled architecture on the substrate is known to contribute to the final material performance (mechanical strength, durability, stability, wettability, efficiency, among others). It is therefore hypothesized that utilizing available cellulose non-wovens instead of electrospun substrates will synergistically combine the ease of processing, scalability, low cost, and the surface functionality rendered by the polysaccharide nanoparticles. Improvements in cost-effectiveness can be expected due to the low amounts of nano-polysaccharide materials needed for surface functionalization. The present work leads to a simple fully bio-based modification of commercially available cellulose non-woven fabrics, which can be used as microfiltration membranes for water purification. Cast coating technique was used to perform the surface modification on the fibers of the nonwoven fabrics. An extensive characterization of the surface morphology, membrane performance, particle rejection, mechanical properties, and antifouling properties were evaluated and discussed in this context. The purpose of the proposed modification concept is to obtain more efficient materials, cost-effective and environmentally friendly than the currently available products; providing sustainable solutions for the upscaling of membrane processes and their integration into water recycling from domestic and industrial effluents.

## 2. Materials and Methods

### 2.1. Materials

One hundred percent cellulose non-woven fabrics were provided by Tekstina, Ajdovscina, Slovenia. CNC 5 wt% in water were purchased from CelluForce, Windsor, QC, Canada; with a crystal diameter of 5–8 nm and lengths of 150–200 nm, determined by atomic force microscopy (AFM) and showed elsewhere [9]). T-CNF (1.1 mmol/g of carboxyl groups) 1.4 wt% in water were prepared following the method reported by Isogai [14] and were provided by Swiss Federal Laboratories of Materials Science and Technology (EMPA), Dübendorf, Switzerland; with a nanofiber diameter 3–5 nm and length in the µm scale, as reported elsewhere [9]. ChNC were prepared following the procedure explained in the Appendix A Chitin nanocrystals preparation process, and then characterized by AFM imaging, (See Appendix A). Polystyrene (PS) latex microspheres were purchased from Alfa Aesar, Kandel, Germany with particle size 50 nm (2.5 wt% dispersion in water), 500 nm (2.5 wt% dispersion in water) and 2 µm (10 wt% dispersion in water). Bovine Serum Albumin (BSA) lyophilized powder ≥96%, phosphate buffered saline (PBS) pH 7.2, and coarse flakes of α-chitin from shrimp shells were purchased from Sigma Aldrich, Stockholm, Sweden. Live/Dead BacLight Bacterial Viability Kit (Molecular Probes, Invitrogen Detection Technologies, Carlsbad, CA, USA), were used as received. The model bacterial strain used for the antibacterial experiments was *Escherichia coli* (*E. coli*) ATCC 8739 (American Type Culture Collection, Manassas, VA, USA).

### 2.2. Processing Method

Non-woven cellulose fabrics were cast coated with CNC, T-CNF, and ChNC suspensions at a speed of 40 mm/s with a gap distance of 150 µm using an automatic film applicator (Elcometer 4340 motorised). Afterwards, the coated fabrics were air-dried for 24 h and dried for 20 min at 100 °C to ensure binding between the nanocrystals/nanofibrils and the cellulose nonwoven fabrics.

### 2.3. Characterization

#### 2.3.1. Morphology and Surface Characterization

The morphology and cross-section of the samples were examined by scanning electron microscopy (SEM) conducted on a JEOL 7000 with an acceleration voltage of 3 kV. The specimens were coated with a thin gold layer prior to visualization using a JEOL JFC-1200 Fine coater at 10 mA for 80 s.

The wettability of the unmodified and coated fabrics was measured and calculated, following the sessile drop technique. An optical contact angle meter from KSV instruments model CAM 200, equipped with a Basler A602f camera was used. Measurements were performed in a conditioned room at 23 °C and relative humidity (RH) of 40 ± 5% RH. For this measurement, the samples were cut and placed on the test cell. Drops of purified water were gently deposited on the sample surface by the delivering syringe. Three water contact angle measurements on each membrane surface were taken at different positions on the sample.

Surface zeta potential (ζ-potential) was measured via electrophoretic light scattering (Zetasizer Nano ZS) using the Surface Zeta Potential Cell (ZEN 1020) from Malvern. Measurements were performed at 25 °C using 10 mM KCl, aqueous solution pH 7.0, with 0.5 wt% poly(acrylic acid) (450 kDa), for negatively charged membranes, and 0.5 wt% polyethylenimine (600 Da), for positively charged membranes, used as tracers. pH was adjusted using 1 M KOH or 1 M HCl.

#### 2.3.2. Membrane Performance

The maximum pore size was determined following ASTM F316–03 Standard Test Methods for Pore Size Characteristics of Membrane Filters by Bubble Point and Mean Flow Pore Test. The samples were pre-wetted for one minute before testing. The test was performed by applying a constant gas pressure using a Sterlitech in dead-end cell mode.

Water permeance of the membranes was measured by filtering deionized water using a Convergence Inspector Titan cross flow equipment with a constant pressure of 0.2 bar for one hour.

The microfiltration performance of the membranes was assessed by studying the particle retention performance. For this, 20 mL of 50 nm, 500 nm, and 2 µm of PS latex microsphere suspensions were passed through the membranes at an applied constant pressure of 0.2 bar. The intensities of light scattered by the feed and permeate solutions were measured via Dynamic Light Scattering (DLS) using the Zetasizer Nano ZS from Malvern Instruments). The parameter *derived counted rate* was used to find the difference on concentration between the feed and permeate, and calculate the rejection rate.

The tensile properties of the membranes were characterized using an Instron 5966 Dual Column Tabletop Testing System equipped with a 100 N load cell. Samples were prepared by cutting rectangular strips of 10 mm × 100 mm from the fabrics. The average sample thickness was measured and taken into account for the calculation of the stress. Prior to the test, the samples were conditioned for 40 h at 50 ± 5% RH and 23 ± 2 °C. The tensile test was performed at a speed of 25 mm/min until failure.

Wet tensile mechanical properties were measured with the same parameters but under wet conditions using a BioPuls Temperature controlled bath to simulate real usage conditions. Samples were pre-conditioned by placing them underwater, held by the grips for 5 min before starting the test. The temperature of the water bath was kept at 25 ± 2 °C.

#### 2.3.3. Antifouling Performance

The antifouling performance of the fabrics was analyzed based on organic fouling and biofouling. First, the antifouling behavior of the modified fabrics was tested for proteins as model for organic foulants, using bovine serum albumin (BSA). The antibacterial properties of the fabrics were assessed using *Escherichia coli* (*E. coli*) ATCC 8739 strain.

##### Determination of Bovine Serum Albumin Protein Adsorption

BSA adsorption was performed using PBS as a buffer solution. The proteins were dissolved in PBS solution at a concentration of 1 mg/mL. The fabrics were equilibrated with PBS overnight and then immersed in the protein solution for 6 h at 37 °C. After that, the fabrics were rinsed three times with PBS solution. In one batch of the samples, the absorbed proteins were removed by immersing the fabrics in 1 wt% sodium dodecyl sulfate (SDS) solution for 1 h at 37 °C under slight shaking conditions. The SDS solution used for rinsing the fabrics was analyzed using a UV-vis spectrophotometer (Genesys 150). In the other batch of the samples, the proteins were kept on the fabrics for their confocal microscopy visualization to study the antifouling properties of the membranes. The proteins were visualized using a Qubit Protein Assay Kit and with a Zeiss LSM 780 Confocal fluorescence microscope (Carl Zeiss MicroImaging GmbH, Oberkochen, Germany). For BSA visualization, the excitation/emission wavelengths were 485 nm and 592 nm respectively.

##### Bacterial Colonization and Viability

Bacterial colonization was assessed after incubation of the different samples with bacterial cells on polystyrene 24-well plates. Exponentially growing cultures of *Escherichia coli* (*E. coli*) ATCC 8739 on nutrient broth (NB) were diluted to an optical density of 0.0138 at a wavelength of 600 nm (OD_600_), equivalent to 10^8^ cells/mL. Two milliliters of diluted cultures were placed on the surface of the fabrics, which were subsequently incubated for 18 h at 37 °C without stirring. After the biofilm assay, the liquid culture was removed and fabrics were carefully rinsed with distilled water to remove not adhered cells.

To evaluate bacterial adhesion and viability, the Live/Dead BacLight Bacterial Viability Kit (Molecular Probes, Invitrogen Detection Technologies, Carlsbad, CA, USA) was used. After bacterial incubation, with the unmodified and coated fabrics for 18 h at 37 °C, the samples were stained with 10 µL BacLight stain (a mixture of SYTO 9 and Propidium Iodide, PI, in DMSO) according to the manufacturer’s recommendations, and incubated in the dark for 15 min at room temperature. All images were acquired using a Leica Microsystems Confocal SP5 fluorescence microscope (Leica Microsystems, Wetzlar, Germany). For green fluorescence (SYTO 9, live cells), excitation was performed at 488 nm (Ar) and emission was recorded at 500–575 nm. For red fluorescence (PI, dead cells), the excitation/emission wavelengths were 561 nm (He-Ne) and 570–620 nm, respectively.

## 3. Results and Discussion

### 3.1. Surface Morphology

The coating parameters were optimized based on the flux values and the stability of the modification, see Appendix A. The selected parameters, shown in this work, were a concentration of 1 wt% nano-polysaccharide suspension, 40 mm/s coated speed, and 150 µm gap distance. SEM micrographs of the unmodified cellulose-based nonwoven fabrics and the CNC, T-CNF, and ChNC coated fabrics are shown in Figure 1. In general, the nanocrystals/nanofibrils suspensions were successfully and homogeneously impregnated on the surface of the fibers. Impregnation of the nanocrystals onto the surface of the non-woven fibers was desirable to ensure their surface functionality and accessibility. In addition, they were considered to be stabilized via H-bonding created during the drying step. Figure 1b–d shows the build-up and film formation tendency of the nanocrystals/nanofibrils in the junctions (crossover) of the non-woven fibers, especially prominent in the cases of T-CNF (Figure 1c). Figure 1f–h also shows in detail the nanotextured surface, achieved on the surface of the fibers after coating them with nanocrystals/nanofibrils, compared to the smooth surface of the unmodified fibers (Figure 1e).

Unmodified cellulose nonwoven fabrics had a hydrophobic contact angle of 106.37 ± 2.73°, whereas for coated fabrics the values were much lower reaching 68.24 ± 3.68° for CNC, 49.24 ± 1.08° for T-CNF, and extremely hydrophilic (contact angle of 0°) for ChNC impregnated samples, since all the water drop was immediately absorbed by the fabric (Table 1). The contact angles are dependent upon the chemical composition, among other factors, and hydrophilicity increases with the presence of N, O, I, Cl, H, and F. Thus, the three types of surface modifications tested in this work, increased the hydrophilicity of the fabrics due to the abundant hydroxyl groups (CNC and T-CNF) and N (ChNC) in their structure [9,15,16,17].

Table 1 also shows the surface ζ-potential for the studied samples at pH 7. All the fabrics were negatively charged, except for the ChNC impregnated ones that were positively charged at this specific pH value. The fabrics impregnated with CNC and T-CNF presented slightly more negative values than the unmodified ones. In the case of CNC and T-CNF impregnated fabrics, surface ζ-potential values were explained by the presence of sulfate half-ester groups or carboxyl groups in the nanocrystal or nanofiber structure, respectively [18,19]. Chitin nanocrystals possess amino groups due to acid hydrolysis-induced deacetylation, the protonation of which makes the surface overall less negative or even positively charged [20].

### 3.2. Membrane Performance

The maximum pore size of the different non-woven fabrics was studied using the bubble point method. As expected, the surface modification done to the original fabrics slightly decreased the pore size, around 7% of the original maximum size for the T-CNF impregnated fabrics, and 6% and 5% for ChNC and CNC, respectively (as shown in Table 1).

The water permeance of the unmodified cellulose-based nonwoven fabrics and the CNC, T-CNF, and ChNC impregnated fabrics are shown in Table 1. CNC and ChNC coated cellulose-based nonwoven membranes showed higher permeance values compared to the unmodified cellulose-based nonwoven fabric. ChNC modified fabrics showed a slight increase in water permeance of 2% (13,417.6 ± 1229.5 L$m^{-2}h^{-1}{\mathrm{bar}}^{-1}$) compared to the unmodified fabrics (13,154.3 ± 2092 L$m^{-2}h^{-1}{\mathrm{bar}}^{-1}$), while CNC presented the best performance in terms of water permeance, with an increase of 9.19% (14,363.8 ± 1228.4 L$m^{-2}h^{-1}{\mathrm{bar}}^{-1}$) respect to the unmodified cellulose nonwoven fabric. However, fabrics impregnated with T-CNF presented a decrease in permeability of 43.3% (7498.4 ± 211.1 L$m^{-2}h^{-1}{\mathrm{bar}}^{-1}$) compared to the unmodified cellulose fabrics. The SEM micrographs and maximum bubble point results showed a reduction of the pore sizes after the coating process due to the web formation between fibers, but at the same time, the intrinsic properties of the nanocrystals or nanofibers improved significantly the wettability of the fabrics making them more hydrophilic, as reflected the water contact angle values in Table 1. A balance between the concentration of the nanomaterial and the thickness of the coating layer over the fabric, together with the values of the wettability, would explain the observed behavior for the high flux obtained in this type of membranes, in agreement with results published elsewhere on the effect of nanocellulose addition to polymeric membranes [21,22,23]. Previous studies showed that cellulose acetate electrospun impregnated with CNCs could reach permeance values of 22,000 L$m^{-2}h^{-1}{\mathrm{bar}}^{-1}$ [12], while commercially available polyethersulfone membranes coated with CNCs presented permeance values of 3180 L$m^{-2}h^{-1}{\mathrm{bar}}^{-1}$ [9]. When comparing the permeance values obtained in this study against previous research, it is possible to understand the great potential of using these commercially available nonwoven fabrics together with the currently proposed coatings as a membrane for water treatment.

The filtration efficiency of the unmodified and impregnated samples was studied for three different particle sizes (50 nm, 500 nm, and 2 μm). For all the systems, the filtration efficiency data showed non-linear trend. In the case of the smallest particles (50 nm), the maximum rejection efficiency was for CNC impregnated fabric with a 12% rejection, and going all the way down to 6% for T-CNF impregnated samples. These low rejection values are related to the difference of sizes between the pore size of the membrane and the particle size. Due to the broad pore size distribution of non-woven membranes, the pore size of the membrane in some areas is at least one order of magnitude larger than the particles, so most of the smallest particles will travel through paths where they can freely flow without being retained on the membrane.

Higher filtration efficiency were observed for the 500 nm particles, showing values as high as 93% for CNC impregnated samples. If the size of the particles is comparable or smaller than the pore size, the particles can flow into the membrane and get retained into settling zones deep into the membrane. This phenomenon is observed on the SEM images (Figure 2f–i). We suggest that the 500 nm particles have higher chances to get retained into settling zones through the membrane since they are not expected to get retained on the surface (compared to bigger sized particles). Although multiple parameters affect the filtration efficiency, this behavior could be mainly attributed to the pore size and pore size distribution of non-woven fabrics. The non-uniform structure of the cellulose membrane causes a tortuous flow path [24], which in turn, causes multiple mechanisms by which particles are retained on the membrane.

For membranes exposed to 2 µm particles, the rejection was improved only for the T-CNF impregnated fabrics. We hypothesize that in this case, the beads were not only attached to the fibers but were also retained on the surface by the web-like structure formed by the coating (Figure 2c,d). This rejection performance could work for removing bigger size particles, such as microplastics from wastewater [25]. For the unmodified, CNC, and ChNC impregnated ones, it is possible to see that the beads only get attached to the fibers. Nevertheless, the permeate results showed that the rejection was lower for CNC and ChNC, compared to the unmodified ones. These lower values of rejection for the ChNC samples could be attributed to the increase in the hydrophilicity of the surfaces after the modification, as demonstrated in the contact angle values from Table 1. The PS beads used in this test are highly hydrophobic, which avoid them from adhering so easily to a hydrophilic surface. In the case of the CNC impregnated fabrics, the lack of web-like structures could be the responsible for this result. It is important to mention that a high standard deviation is observed for measurements of 2 μm particles. This could be caused for the multiple scattering generated by micron-size particles, which could affect the accuracy of the intensity measured by the particle analyzer.

As previously mentioned, there are multiple mechanisms involved in the filtration efficiency of micron and submicron particles, and a full understanding of the influence of each factor is outside of the scope of this work. Nevertheless, our results demonstrate that the modified non-woven fabrics could be used to separate solutes of the size of bacteria and viruses through size exclusion mechanism [26]. Figure 2n summarizes the particle rejection experiment.

The tensile properties of cellulose-based nonwoven samples were measured at dry and wet conditions and are summarized in Table 2 and Figure 3a. Nonwoven fabrics consist of randomly oriented fibers (see Figure 3b) and their mechanical properties depend on their direction and the interaction between fibers. The results at dry conditions showed that the addition of the nanocrystals or nanofibrils coatings, in general, increased the maximum tensile strength and E-modulus, therefore, the stiffness of the unmodified cellulose nonwoven fabrics, but the elongation at break decreased. The behavior of the T-CNF impregnated fabric deserves special attention; it not only resulted in greater values of tensile strength but also considerably increased the E-modulus compared to the rest of the samples. This higher reinforcement effect, compared to CNC and ChNC at the same concentration, could be attributed to its higher aspect ratio [27], leading to the formation of secondary ultrafine structures interacting with primary nonwoven fibers through bonding points and to a better stress transfer favored by the T-CNF nanofibers’ alignment [28]. This is supported by the SEM images from Figure 1c, where it is possible to see that the T-CNF impregnated fabrics formed more web-like structures within the fibers. In wet conditions, similar to the ones the membranes will be exposed during their real application, both tensile strength and E-modulus of the impregnated fabrics were drastically reduced compared to the dry conditions value, neutralizing almost completely the reinforcement effect from the different modifications. When cellulose fabrics are exposed to humid environments, they absorb water and it acts as a plasticizer. This results in less stiffness and sometimes a higher elongation at break. In this case, the elongation at break values were not consistent to the dry behavior and showed the same or even lower values than in those conditions. It is important to mention that nonwoven fibers are randomly aligned and will break at different rates. Some of them might remain holding the structure for a longer time or slide from the nonwoven structure, showing a different mechanism to regular solid material breaks. Furthermore, none of the studied samples formed a continuous layer on top of the fabric, but were impregnated over the individual fibers or generated bonding points between the fibers, therefore most of the mechanical properties should be dictated by the properties of the support, in this case, the unmodified nonwoven fabric.

### 3.3. Antifouling Performance

#### 3.3.1. Bovine Serum Albumin Adsorption

Understanding protein adsorption to surfaces is of importance for various environmental and biomedical applications. Many bio-related responses such as biofilm formation and cell adhesion depend on several factors in their initial steps. Since protein adsorption is believed to be one of the first events occurring in such responses [29], we considered static bovine serum albumin adsorption as a primary method to evaluate the antifouling properties of the impregnated nonwoven fabrics. Adsorption of BSA to the modified and unmodified surfaces was investigated using UV-vis spectroscopy and confocal microscopy in a complementary way.

Figure 4A shows the adhesion of BSA on the surface of the unmodified and coated fabrics, whereas Table 3 summarizes the recovery of the non-adsorbed protein from the surface of the membranes in SDS 1%, measured as the BSA absorbance at 280 nm. The staining of BSA with Qubit Protein Assay Kit revealed that the highest adsorption of protein, at pH 7, was on the ChNC impregnated nonwoven fabric (Figure 4d), followed by the unmodified fabric (Figure 4a), the CNC impregnated fabric (Figure 4b), and the lowest adsorption was observed on the T-CNF impregnated fabric (Figure 4c). Data obtained with UV-vis spectroscopy (Table 3) show the lowest BSA recovery in SDS from the ChNC impregnated fabric after 6 h of incubation with the protein solution. Specifically, this value represents 4.66% compared to the proteins recovered from the T-CNF impregnated fabric, followed by the unmodified fabric (11.66% BSA recovery from the surface) and the CNC impregnated fabric (41.33% BSA recovery from the surface). These results are in agreement with the ones observed with confocal microscopy, indicating that lower recovery in SDS means higher adhesion of proteins on the membrane surface, hence, confirming that the protein adsorption was significantly reduced upon T-CNF impregnation.

Generally, protein adsorption at the solid/liquid interface can occur due to electrostatic interactions, hydrophobic interactions, and hydrogen-bonding interactions. This process is influenced by properties of the protein such as stability, the adsorbent surface, and solution such as ionic strength and pH value [30]. Charged surfaces tend to exhibit a greater quantity of bovine serum albumin adsorption, a larger bovine serum albumin layer thickness, and increased density of protein compared to neutral surfaces at neutral pH value [31]. Solution pH affects BSA adsorption as the isoelectric point (IEP) of BSA is at pH 4.5–5.0, therefore the protein is negatively charged at neutral pH and positively charged under acidic conditions [31,32,33,34,35,36,37]. This fact explains the results obtained in this work since the incubation of the fabrics with the BSA solution was performed at pH 7. Fabrics were also incubated with BSA at different pH ranges and BSA presented no adhesion to ChNC impregnated fabrics at pH 3 since both BSA protein and ChNC impregnated surface were positively charged under acidic conditions (data not shown). When BSA is charged negatively, it is partially repelled from the negatively charged surfaces of T-CNF, CNC, and cellulose nonwoven fabrics (Table 1). Therefore, a decrease in BSA adsorption is expected. In the same way, the large BSA adsorption to positive charged surfaces, such as ChNC impregnated fabrics, can be attributed to the strong electrostatic interactions between chitin and BSA negatively charged at neutral pH. Previous studies evidenced by combinatorial quartz crystal microbalance (QCM-D) as a sensitive analytical tool to evaluate attachment and detachment of adsorbed proteins, that after BSA equilibrium sorption is reached at pH 7.0, desorption of bovine serum albumin occurs when the environment pH is below the isoelectric point of the protein [31]. Based on this behavior, the chitin-impregnated fabrics presented in this work could be promising when designing self-cleaning systems with environmental implications.

Hydrophobic interactions and surface chemistry should be also considered to explain protein adsorption. Several studies concluded that the quantity of BSA adsorption was found to decrease in the order –NH_2_ ≥ –COOH > –CH_3_ > –OH and greater BSA adsorption was observed on hydrophobic surfaces [31,38,39]. This is in agreement with the chemical composition of the nanocrystal suspensions used in our coatings, being BSA adsorption higher on ChNC coated samples (–NH_2_ groups). The reduction in protein adsorption on T-CNF and CNC could be also attributed to the abundant hydroxyl (–OH) and carboxyl (–COOH) groups for T-CNF and hydroxyl (–OH) for CNC on their surface; which allows a layer of water to bind to the surface preventing the adhesion of BSA to the surface. In addition, the presence of sulfate half-ester groups (–OSO_3_–) [19] on the surface of the CNCs generates an electrostatical repulsion with the BSA at the pH 7.0. Cellulose presents 3 liable –OH groups (per monomer) [40]. However, BSA adsorption to cellulose uncoated membranes seems to be driven by electrostatic interactions together with its hydrophobic nature more than by its chemical composition based on the results obtained in this work.

#### 3.3.2. *Escherichia coli* Colonization and Cellular Viability

Bacterial colonization is a complex process that involves bacterial adhesion to the substrate and continues with the formation of biofilms. Biofilms are communities of microorganisms adhering to surfaces, which are embedded by a self-produced extra-polymeric matrix aimed at facilitating their survival in adverse environments. In water and wastewater treatment facilities, biofilms, fouling, and biofouling are ubiquitous, with decreased quality of treated water and reduced efficacy of filtration systems. Membrane fouling causes a significant decrease in the permeation flux, which results in substantial increases in energy demand and operational- and maintenance costs [41,42,43,44]. Due to the adverse impact of biofilms, different physical and chemical methods have been investigated to prevent and remove them, particularly in the case of new nanostructured surfaces. In this work, and similarly to BSA adsorption, we focused our studies towards the effect of CNC, T-CNF, and ChNC on bacterial viability and bacterial colonization, using *E. coli* ATCC 8739 as a bacterial strain.

Live/dead double staining was used for monitoring the bacterial viability as a function of the membrane integrity of the cell. Figure 4B shows confocal microscopy micrographs of the presence and viability of *E. coli* on the surface of the non-woven fabrics. In general, all the samples displayed important resistance to be colonized by *E. coli*, as can be deduced from the absence of a large number of bacteria accumulated on the surface of the nonwoven fabrics, being slightly higher on the ChNC impregnated surfaces. Bacterial attachment and biofilm formation are complex processes still poorly understood that depend on several factors, including the physicochemical properties of the surface, the temperature and pH, the availability of nutrients, and the type of strain. However, bio-surface interactions are more complex, and other factors such as surface charge and hydrophobicity have been shown to influence microbial attachment [45]. Several studies reported that microorganisms preferably attach to hydrophobic nonpolar surfaces rather than to hydrophilic materials [46,47,48]. Other studies revealed that bacterial colonization took place preferentially on membranes with intermediate hydrophilicity values, whereas super hydrophobic and super hydrophilic surfaces presented lower affinity for bacteria [49,50]. Our results showed that CNC and T-CNF impregnated fabrics displayed moderate hydrophilicity compared to the high hydrophobic unmodified cellulose fabrics and moderate hydrophobicity compared to the highly hydrophilic ChNC impregnated ones (Table 1). Nevertheless, other important factor governing bacterial adhesion is surface charge. Given the negative surface charge of bacterial outer membranes (the ζ-potential of *E. coli* is about −30 mV) [48] and the negative surface charge of all nonwoven fabrics in this study, except those impregnated with ChNC (Table 1); electrostatic repulsions could explain the low bacterial colonization observed on the unmodified, CNC and T-CNF impregnated nonwoven fabrics (Figure 4e,f,g, respectively). On the other hand, the surface of the ChNC impregnated fabrics was charged positively, which could interact with the negatively charged outer membrane of the bacteria promoting a higher adhesion compared to CNC and T-CNF impregnated ones, explaining results like those obtained in this work. Nonetheless, the rationalization of bacterial attachment to surfaces exclusively in terms of their physio-chemical characteristics has low predictive value for at least two reasons. First, cell binding is affected by the culture media used due to differences in surface tension or the absorption of organic and inorganic compounds, which modify the way microorganisms adhere [46]. Second, bacterial morphology makes cell-surface interactions a complex issue due to the existence of cell appendages and adhesion structures avoiding direct contact [51]. A multidisciplinary approach that considers not only the relationship between hydrophobicity and surface charge, but also its topography as well as the interactions of the bacterial appendages with the surface of the fabrics would be the key to have a better explanation of how, in this case, bacteria adhere to these types of cellulose-based nanomaterials. Regarding bacterial viability, the SYTO9/PI double staining reveals cells with a compromised fabric that are considered to be dead or dying will stain red, whereas cells with an intact membrane will stain green. Figure 4B shows that either unmodified cellulose nonwoven fabrics (Figure 4e) or CNC coated ones (Figure 4f) did not significantly impair bacterial cells, as noted by the absence of red marked (cell membrane-damaged) bacteria. Conversely, T-CNF and ChNC impregnated nonwoven fabrics remarkably reduced the viability of the cells on their membrane surface, as shown by the high number of red or yellow marked non-viable cells (Figure 4g,h, respectively), being this antibacterial effect more evident in the case of the ChNC coating. It is well known that microorganisms have certain general physiological requirements for survival and growth. These include an appropriate temperature range, nutrients, oxygen (or lack thereof, for anaerobic bacteria), moisture, and pH. Extremes of acidity or alkalinity can effectively limit the growth and survival of microorganisms, and the presence of carboxyl groups in T-CNF [41], can provide enough antibacterial activity through a change of pH in the bacterial environment, lowering it [52,53], hence making it hostile for bacterial survival. The antibacterial effect of natural chitin is believed to arise from a small portion of deacetylated structural units in their chitin structure [6]. The acid hydrolysis produced during the extraction of chitin nanocrystals enlarges the proportion of deacetylated groups with the outcome of high antibacterial activity [54,55]. The hydrolytic treatment leads to the formation of NH_3_^+^ groups, which interact with the negatively charged residues of carbohydrates, lipids, and proteins located on the cell surface of bacteria, so explaining their role in bacterial impairment [56,57]. Previous studies support the model in which the electrostatic forces between protonated –NH_3_^+^ groups and the negative residues mediated the interaction, presumably by competing with Ca_2_^+^ for electronegative sites on the membrane surface [7,49]. However, further studies at the molecular level need to be done in order to have a better understanding of the antibacterial activity of chitin.

## 4. Conclusions

Cellulose non-woven fabrics were successfully developed into fully bio-based membranes for water purification, by impregnating different nanopolysaccharide (CNC, T-CNF, or ChNC) by cast coating technique. This resulted in a nanotextured surface on textile fibers. The nanopolysaccharide coatings improved significantly the hydrophilicity/wettability of the fabric and altered the surface charge; being more negative in the case of cellulose-based modifications, or positive in the case of chitin. CNC and ChNC modifications improved the permeance of the membranes. The nanocrystals or nanofibers impregnation on the original fabrics increased mechanical properties (the tensile strength and E-modulus); with T-CNF impregnated ones showing drastically improved stiffness in dry conditions.

T-CNF modified fabrics successfully separate the 2 µm particles, showing a potential to be used for microplastics filtration. Furthermore, all of the systems showed excellent separation properties within a range size of 500 nm, which demonstrated that they can be a suitable option for separating bacteria and some viruses. It may be noted that we have evaluated the effect of pore structure on the filtration efficiency using uncharged PS beads and separation due to charge is not reported here. The coated fabrics further showed antifouling capability against proteins and T-CNF and ChNC impregnations proved to be harmful to the bacteria colonizing the surface of the fabrics.

All of the modifications showed a great potential of using commercially available nonwoven fabrics together with the currently proposed coatings as a membrane for water treatment, due to the easy processing, enhanced performance, reduced energy cost, and longer service life which is expected to contribute to a new generation of scalable biobased microfiltration membranes for environmental applications.

## Figures and Tables

**Figure 1 nanomaterials-11-01752-f001:**
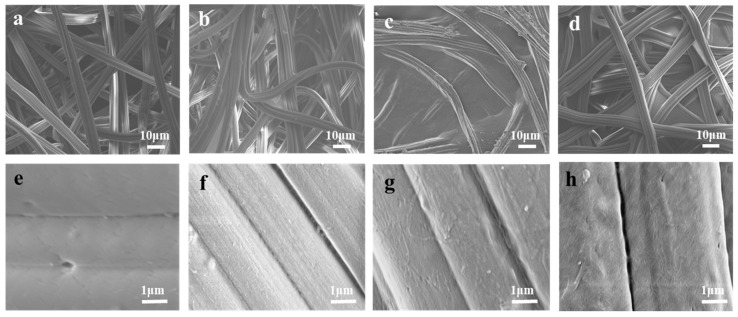
SEM images of unmodified and nonwoven fabrics after the coating process. (**a**) Surface of cellulose unmodified fabric, (**b**) surface of CNC impregnated fabric, (**c**) surface of T-CNF impregnated fabric, (**d**) surface of ChNC impregnated fabric, (**e**) surface of cellulose unmodified fibers, (**f**) nanotextured surface of CNC impregnated fibers, (**g**) nanotextured surface of T-CNF impregnated fibers, and (**h**) nanotextured surface of ChNC impregnated fibers.

**Figure 2 nanomaterials-11-01752-f002:**
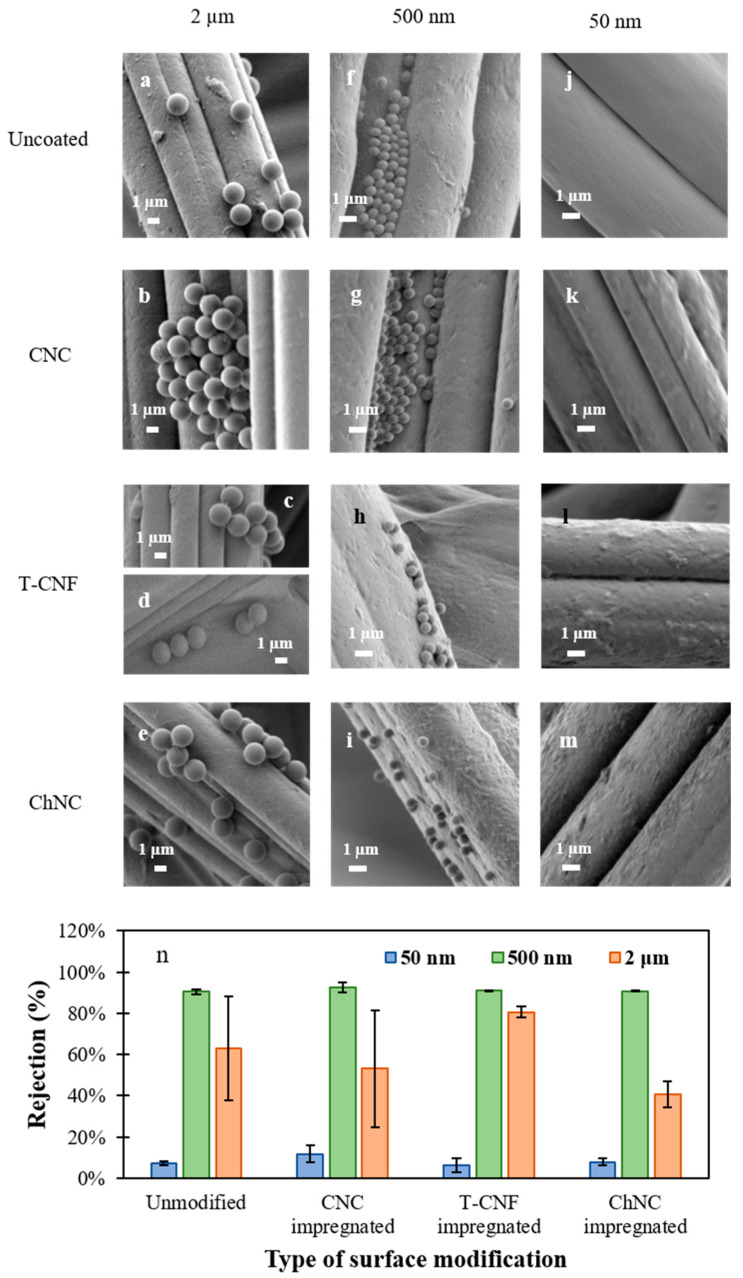
SEM images of unmodified and nonwoven fabrics after coating process, used for the particle retention performance study. For 2 µm PS beads suspension: (**a**) unmodified nonwoven fabric, (**b**) CNC impregnated, (**c**,**d**) T-CNF impregnated, (**e**) ChNC impregnated; for 500 nm PS beads suspension: (**f**) unmodified nonwoven fabric, (**g**) CNC impregnated, (**h**) T-CNF impregnated, (**i**) ChNC impregnated; for 50 nm PS beads suspension: (**j**) unmodified nonwoven fabric, (**k**) CNC impregnated, (**l**) T-CNF impregnated, (**m**) ChNC impregnated, and (**n**) summary of rejection values for 50 nm, 500 nm and 2 µm particles.

**Figure 3 nanomaterials-11-01752-f003:**
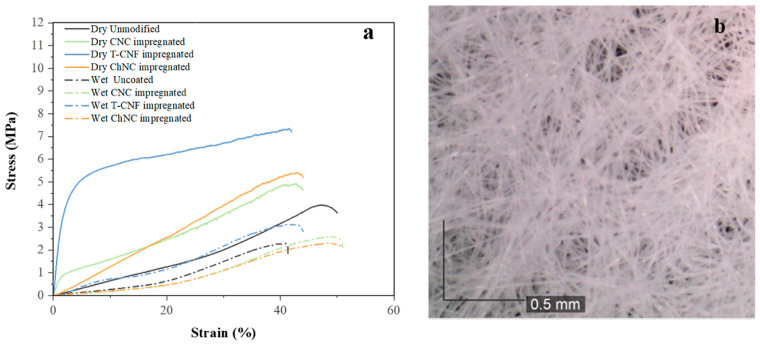
(**a**) Stress–strain curves in dry and wet conditions of unmodified, CNC impregnated, T-CNF impregnated and ChNC impregnated fabrics, and (**b**) random orientation of nonwoven fibers.

**Figure 4 nanomaterials-11-01752-f004:**
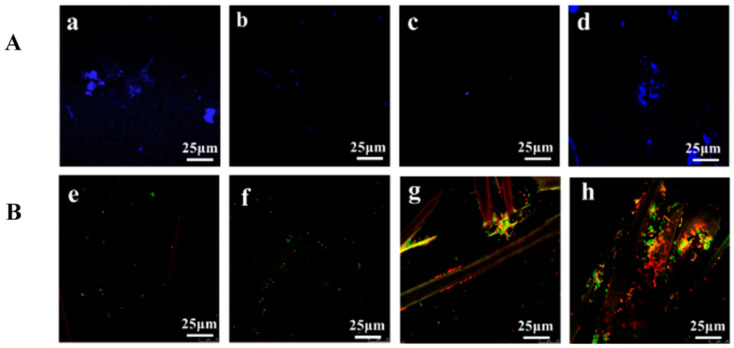
(**A**) Confocal microscopy micrographs showing BSA adsorption on the surface of the unmodified (**a**), CNC impregnated (**b**), T-CNF impregnated (**c**) and ChNC impregnated (**d**) nonwoven fabrics stained with the Qubit Protein Assay Kit after their incubation with BSA solution for 6 h at pH 7. (**B**) Confocal microscopy micrographs showing *E. coli* colonization and viability on the surface of the unmodified (**e**), CNC impregnated (**f**), T-CNF impregnated (**g**) and ChNC impregnated (**h**) nonwoven fabrics stained with Live/Dead BacLight Bacterial Viability Kit. Live cells were green stained by SYTO 9 and dead cells were red-stained by PI.

**Table 1 nanomaterials-11-01752-t001:** Contact angle, surface ζ-potential, maximum pore size, and permeance of unmodified and impregnated fabrics.

Sample	Water Contact Angle (°)	Surface ζ-Potential (pH 7, mV)	Maximum Pore size (µm)	Water Permeance (L$m^{-2}h^{-1}bar^{-1})$
Cellulose unmodified membrane	106.37 ± 2.7	−2.6 ± 1.3	2.11 ± 0.18	13,154.3 ± 2092
CNC impregnated membrane	68.24 ± 3.7	−5.9 ± 2.8	2.00 ± 0.15	14,363.8 ± 1228.4
T-CNF impregnated membrane	49.24 ± 1.1	−9.2 ± 2.5	1.96 ± 0.13	7498.4 ± 211.1
ChNC impregnated membrane	0 ^a^	7.2 ± 3.2	1.98 ± 0.10	13,417.6 ± 1229.5

^a^ Too low to be measured.

**Table 2 nanomaterials-11-01752-t002:** Summary of mechanical properties for uncoated and impregnated membranes.

Conditions	Sample Type	Maximum Tensile strength (MPa)	E-Modulus (MPa)	Elongation at Break (%)
Dry	Unmodified cellulose fabric	4.3 ± 0.5	9.0 ± 1.6	51 ± 2
	CNC impregnated fabric	4.8 ± 0.2	46.9 ± 0.5	44 ± 1
	T-CNF impregnated fabric	7.2 ± 0.2	194.8 ± 0.5	39 ± 3
	ChNC impregnated fabric	5.1 ± 0.4	12.3± 1.0	45 ± 2
Wet	Unmodified cellulose fabric	2.2 ± 0.1	8.2 ± 0.4	44 ± 2
	CNC impregnated fabric	2.6 ± 0.0	8.5 ± 0.4	48 ± 2
	T-CNF impregnated fabric	3.00 ± 0.1	8.8 ± 0.6	44 ± 1
	ChNC impregnated fabric	2.2 ± 0.1	7.6 ± 0.3	49 ± 1

**Table 3 nanomaterials-11-01752-t003:** UV-vis absorbance of BSA at 280 nm for unmodified, CNC impregnated, T-CNF impregnated and ChNC impregnated nonwoven fabrics exposed to BSA solution for 6 h at pH 7.

Sample	UV-vis Absorbance at 280 nm
Unmodified cellulose fabric	0.04 ± 0.04
CNC impregnated fabric	0.12 ± 0.12
T-CNF impregnated fabric	0.30 ± 0.17
ChNC impregnated fabric	0.014 ± 0.02

## Data Availability

The data presented in this study are available within the article and in the Appendix A.

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
