# Peer review of "Water Filtration Membranes Based on Non-Woven Cellulose Fabrics: Effect of Nanopolysaccharide Coatings on Selective Particle Rejection, Antifouling, and Antibacterial Properties"

_nanomaterials, 2021, doi:10.3390/nano11071752_

Round 1
Reviewer 1 Report
Comment for nanomaterials-1278049 is listed as follows,
1. There are some miss been named or error typing.
(1) In p13 and p15, please check the subtitles no. of "3.4.1", they are duplicated. Also check the subtitles no. of "3.3", they are missed.
(2) In p1, please check "micro (0.1-100 m)", please provide the abbreviation for "IUPAC".
(3) In p2, please change "microfiltration (MF) [2]" into " MF [2]".
(4) In p3, please change "Cellulose 87 nanocrystals (CNC) 5 wt% in" into "CNC 5 wt% in".
(5) In p2 and p3, please check "TEMPO-oxidized cellulose nanofibrils (T-CNF) " with "TEMPO cellulose nanofibers (T-CNF, …)" they are in same abbreviations but different name.
(6) In p3, please change " Chitin nanocrystals (ChNC)" into " ChNC".
(7) In p5, please check "0.01 M PBS".
(8) In p17, please check "-T-CNF".
(9) In Figure 3a, the curve of "dry T-CNF impregnated fabric" is interrupted by the symbols, the size of symbols should be reduced.
(10) In Table 2, please check the E-modulus (MPa) number of "dry T-CNF impregnated fabric".
Author Response
Top of Form
Journal: Nanomaterials (ISSN 2079-4991)
Manuscript ID: nanomaterials-1278049
Type: Article
Number of Pages: 18
Title: Water filtration membranes with selective particle rejection, antifouling and antibacterial properties based on non-woven cellulose fabrics coated with nanopolysaccharides
Authors: Blanca Jalvo, Andrea Aguilar-Sanchez, Maria Ximena Ruiz Caldas, Aji P. Mathew *
We would like to thank the editor and reviewers for their positive feedback and constructive comments/suggestions. A point by point response to reviewer comments are provided below in blue font. As suggested, the following changes have been incorporated in the revised manuscript.
Please note that we have also revised the title to ‘Water filtration membranes based on non-woven cellulose fabrics: Effect of nanopolysaccharide coatings on selective particle rejection, antifouling and antibacterial properties’
Reviewer #1
- There are some miss been named or error typing.
(1)In p13 and p15, please check the subtitles no. of "3.4.1", they are duplicated. Also check the subtitles no. of "3.3", they are missed.
Thanks! This was corrected.
(2) In p1, please check "micro (0.1-100 m)", please provide the abbreviation for "IUPAC".
Thanks! This was corrected and added.
(3) In p2, please change "microfiltration (MF) [2]" into " MF [2]".
Thanks! This was corrected.
(4) In p3, please change "Cellulose 87 nanocrystals (CNC) 5 wt% in" into "CNC 5 wt% in".
Thanks! This was corrected.
(5) In p2 and p3, please check "TEMPO-oxidized cellulose nanofibrils (T-CNF) " with "TEMPO cellulose nanofibers (T-CNF, …)" they are in same abbreviations but different name.
Thanks! This was corrected.
(6) In p3, please change " Chitin nanocrystals (ChNC)" into " ChNC".
Thanks! This was corrected.
(7) In p5, please check "0.01 M PBS".
Thanks! This was corrected.
(8) In p17, please check "-T-CNF".
Thanks! This was corrected.
(9) In Figure 3a, the curve of "dry T-CNF impregnated fabric" is interrupted by the symbols, the size of symbols should be reduced.
Thanks for the observation! This has been modified.
(10) In Table 2, please check the E-modulus (MPa) number of "dry T-CNF impregnated fabric".
Thanks for the comment! We checked and this number is correct. There was a major improvement, which can be observed in Figure 3a.
Bottom of Form
Reviewer 2 Report
"Water filtration membranes with selective particle rejection, antifouling and antibacterial properties based on non-woven cellulose fabrics coated with nanopolysaccharides" is an interesting paper. Some improvement is required!
Line 88: nanocrystals (CNC) 5 wt% in water were purchased from CelluForce, Canada; with crystal diameter of 5-8 mm (is it correct?) and lengths of 150-200 nm
Line 32, 33: The removal Efficiency and selectivity of the filters/Membranes (for metallic Ions from solution) is related..? Please to add it!
Line 290: Currently proposed coatings as a membrane for water treatment (Wastewater Treatment?)
Line 291, 292: The filtration efficiency of the unmodified and impregnated samples was studied for three different particle sizes (50 nm, 500nm, and 2μm). Please to add a type of three different particle sizes.
Line 301: Higher filtration efficiency were observed for the 500 nm particles, showing values as high as 93% for CNC impregnated samples. Is it valid only for spherical particles, or also for cylindrical particles?
Line 335: Can you add shortly the origin (synthesis method) of ideally spherical particles shown at Figure 2.
Conclusion
Line 563: All of the mofications showed..(Please to change it in "modification
Line 564: a great potential of using commercially available
nonwoven fabrics together with the currently proposed coatings as a membrane for water treatment,...
It is not clear, what is relationship of the filtration effficiency of the investigated materials and characteristics of water (pH, zeta potential, electrical conductivity,...). Can you explain it in conclusion!
Author Response
Top of Form
Journal: Nanomaterials (ISSN 2079-4991)
Manuscript ID: nanomaterials-1278049
Type: Article
Number of Pages: 18
Title: Water filtration membranes with selective particle rejection, antifouling and antibacterial properties based on non-woven cellulose fabrics coated with nanopolysaccharides
Authors: Blanca Jalvo, Andrea Aguilar-Sanchez, Maria Ximena Ruiz Caldas, Aji P. Mathew *
We would like to thank the editor and reviewers for their positive feedback and constructive comments/suggestions. A point by point response to reviewer comments are provided below in blue font. As suggested, the following changes have been incorporated in the revised manuscript.
Please note that we have also revised the title to ‘Water filtration membranes based on non-woven cellulose fabrics: Effect of nanopolysaccharide coatings on selective particle rejection, antifouling and antibacterial properties’
Bottom of Form
Reviewer #2
Line 88: nanocrystals (CNC) 5 wt% in water were purchased from CelluForce, Canada; with crystal diameter of 5-8 mm (is it correct?) and lengths of 150-200 nm
Thanks! The correct value is 5-8 nm and it has been corrected in the manuscript already.
Line 32, 33: The removal Efficiency and selectivity of the filters/Membranes (for metallic Ions from solution) is related..? Please to add it!
Thanks for the observation! This is already mentioned a couple of lines later:
The removal efficiency or selectivity of the membrane/filters is related to its microstructure, pore size, and surface chemistry. Nano (0.1-100 nm range), micro (0.1-100 µm), and milli (pore size 0.1-100mm) pores based on International Union of Pure and Applied Chemistry (IUPAC) definition allows the progressive removal of multivalent ions, virus, bacteria, and suspended particles by membranes and filters [1]. In the case of charged membrane as in the current case the removal via electrostatic interactions provide separation independent of size.
We do consider that our coatings can therefore be used for metallic ions, and in general charged molecules since it was showed that CNC and T-CNF modified the surface charge towards more negative values and ChNC toward positive values.
Line 290: Currently proposed coatings as a membrane for water treatment (Wastewater Treatment?)
Thank you for your observation! They can be used for wastewater treatment, but also for other separation processes. This is the reason why we do not narrow it down just to waste water.
Line 291, 292: The filtration efficiency of the unmodified and impregnated samples was studied for three different particle sizes (50 nm, 500nm, and 2μm). Please to add a type of three different particle sizes.
Thanks for the comment. This is already mentioned in lines 98-100 from the 2.1 Materials section.
Polystyrene (PS) latex microspheres were purchased from Alfa Aesar, Germany with particle size 50 nm (2.5wt% dispersion in water), 500 nm (2.5wt% dispersion in water) and 2 µm (10 wt% dispersion in water).
Line 301: Higher filtration efficiency were observed for the 500 nm particles, showing values as high as 93% for CNC impregnated samples. Is it valid only for spherical particles, or also for cylindrical particles?
With the current information is hard to give a straight forward response since it is necessary to analyze the possible transport mechanisms through the membranes. The mechanism of transport observed on these particles is related to the differences between the mean free path of the particle and the pore size. Besides that, when we think about non-spherical particles with long aspect ratios -such as cylinders- there are other mechanisms involved in the flow such as rotational motion. In our case, the dominant transport mechanism is viscous flow since there is a difference in pressure as a driving force for the flow of particles. For example, in the case of the smallest particles, the pore size is significative larger than the mean free path of 50nm particles, so we expect cylindrical particles with a similar hydrodynamical diameter to get a low retention. For the spherical particles around 500nm we hypothesized that the high retention is caused by the particles getting retained into settling zones deep into the membrane. It is known that cylindrical particles are nearly aligned with the flow direction. Hence, in the case of cylindrical particles with a mean free path in the same order of magnitude as the pore size, it may happen that the chances of getting retained are lower, which would decrease the filtration efficiency. However, the cylindrical particles could aggregate through the pores and obstruct them, which would result in an increment of the filtration efficiency.
Due to the lack of clear experimental data on this we prefer not to add any discussion on this to the manuscript.
Line 335: Can you add shortly the origin (synthesis method) of ideally spherical particles shown at Figure 2.
It may be noted that the polystyrene particles used in this study were purchased from Alfa Aesar and are typically used for membrane characterization. It was not prepared by us.
https://www.alfa.com/en/catalog/041918/
https://www.alfa.com/en/catalog/042714/
https://www.alfa.com/en/catalog/042711/
Conclusion
Line 563: All of the mofications showed..(Please to change it in "modification
Thanks! This was corrected.
Line 564: a great potential of using commercially available
nonwoven fabrics together with the currently proposed coatings as a membrane for water treatment,...
It is not clear, what is relationship of the filtration efficiency of the investigated materials and characteristics of water (pH, zeta potential, electrical conductivity,...). Can you explain it in conclusion!
In the current study we have evaluated the filtration efficiency of uncharged PS beads to understand the effect of pore structure and separation due to charge effects were not expected. Furthermore the exact correlation between pH, zeta potential and conductivity is highly complex and cannot be conclude from this study. In the current study the filtration efficiency was evaluated at a specific pH (ph7) was evaluated where the zeta potential and conductivity remain constant. We have now mentioned in the conclusion that ‘’Here we have evaluated the effect of pore structure on the filtration efficiency of uncharged PS beads and separation due to charge effects were not studied or reported.’’